# On the Lower Bound of Minimizing Polyak-Łojasiewicz functions

## Abstract

Polyak-Łojasiewicz (PL) (Polyak, 1963) condition is a weaker condition than the strong convexity but suffices to ensure a global convergence for the Gradient Descent algorithm. In this paper, we study the lower bound of algorithms using first-order oracles to find an approximate optimal solution. We show that any first-order algorithm requires at least $\Omega\left(\frac{L}{\mu}\log\frac{1}{\varepsilon}\right)$ gradient costs to find an $\varepsilon$-approximate optimal solution for a general $L$-smooth function that has an $\mu$-PL constant. This result demonstrates the *optimality* of the Gradient Descent algorithm to minimize smooth PL functions in the sense that there exists a "hard" PL function such that no first-order algorithm can be faster than Gradient Descent when ignoring a numerical constant. In contrast, it is well-known that the momentum technique, e.g. (Nesterov, 2003, chap. 2) can provably accelerate Gradient Descent to $O\left(\sqrt{\frac{L}{\hat{\mu}}}\log\frac{1}{\varepsilon}\right)$ gradient costs for functions that are $L$-smooth and $\hat{\mu}$-strongly convex. Therefore, our result distinguishes the hardness of minimizing a smooth PL function and a smooth strongly convex function as the complexity of the former cannot be improved by any polynomial order in general.

## 1 Introduction

We consider the problem

$$\min_{\mathbf{x}\in\mathbb{R}^d} f(\mathbf{x}), \tag{1}$$

where the function $f$ is $L$-smooth and satisfies the Polyak-Łojasiewicz condition. A function $f$ is said to satisfy the Polyak-Łojasiewicz condition if (2) holds for some $\mu > 0$:

$$\|\nabla f(\mathbf{x})\|^2 \geq 2\mu\left(f(\mathbf{x}) - \inf_{\mathbf{y}\in\mathbb{R}^d} f(\mathbf{y})\right), \quad \forall \mathbf{x}\in\mathbb{R}^d. \tag{2}$$

We refer to (2) as the $\mu$-PL condition and simply denote $\inf_{\mathbf{y}\in\mathbb{R}^d} f(\mathbf{y})$ by $f^*$. The PL condition may be originally introduced by Polyak (Polyak, 1963) and Łojasiewicz (Lojasiewicz, 1963) independently. The PL condition is strictly weaker than strong convexity as one can show that any $\hat{\mu}$-strongly convex function which by definition satisfies:

$$f(\mathbf{x}) \geq f(\mathbf{y}) + \langle\nabla f(\mathbf{y}), \mathbf{x} - \mathbf{y}\rangle + \frac{\hat{\mu}}{2}\|\mathbf{x} - \mathbf{y}\|^2$$

is also $\hat{\mu}$-PL by minimizing both sides with respect to $\mathbf{x}$ (Karimi et al., 2016). However, the PL condition does not even imply convexity. From a geometric view, the PL condition suggests that the sum of the squares of the gradient dominates the optimal function value gap, which implies that any local stationary point is a global minimizer. Because it is relatively easy to obtain an approximate local stationary point by first-order algorithms, the PL condition serves as an ideal and weaker alternative to strong convexity.

In machine learning, the PL condition has received wide attention recently. Lots of models are found to satisfy this condition under different regimes. Examples include, but are not limited to, matrix decomposition and linear neural networks under a specific initialization (Hardt & Ma, 2016; Li et al., 2018), nonlinear neural networks in the so-called neural tangent kernel regime (Liu et al.,

2022), reinforcement learning with linear quadratic regulator (Fazel et al., 2018). Compared with strong convexity, the PL condition is much easier to hold since the reference point in the latter only is a minimum point such that $\mathbf{x}^* = \operatorname{argmin}_{\mathbf{y}} f(\mathbf{y})$, instead of any $\mathbf{y}$ in the domain.

Turning to the theoretic side, it is known (Karimi et al., 2016) that the standard Gradient Descent algorithm admits a linear converge to minimize a $L$-smooth and $\mu$-PL function. To be specific, in order to find an $\varepsilon$-approximate optimal solution $\hat{\mathbf{x}}$ such that $f(\hat{\mathbf{x}}) - f^* \leq \varepsilon$, Gradient Decent needs $O(\frac{L}{\mu} \log \frac{1}{\varepsilon})$ gradient computations. However, it is still not clear whether there exist algorithms that can achieve a provably faster convergence rate. In the optimization community, it is perhaps well-known that the momentum technique, e.g. Nesterov (2003, chap. 2), can provably accelerate Gradient Descent from $O(\frac{L}{\hat{\mu}} \log \frac{1}{\varepsilon})$ to $O\left(\sqrt{\frac{L}{\hat{\mu}}} \log \frac{1}{\varepsilon}\right)$ for functions that are $L$-smooth and $\hat{\mu}$-strongly convex. Even though some works (J Reddi et al., 2016; Lei et al., 2017) have considered accelerations under different settings, probably faster convergence of first-order algorithms for PL functions is still not obtained up to now.

In this paper, we study the first-order complexities to minimize a generic smooth PL function and ask the question:

*"Is the Gradient Decent algorithm (nearly) optimal or can we design a much faster algorithm?"*

We answer the question in the language of min-max lower bound complexity for minimizing the $L$-smooth and $\mu$-PL function class. We analyze the worst complexity of minimizing any function that belongs to the class using first-order algorithms. Excitingly, we construct a hard instance function showing that any first-order algorithm requires at least $\Omega\left(\frac{L}{\mu} \log \frac{1}{\varepsilon}\right)$ gradient costs to find an $\varepsilon$-approximate optimal solution. This answers the aforementioned question in an explicit way: the Gradient Descent algorithm is already *optimal* in the sense that no first-order algorithm can achieve a provably faster convergence rate in general ignoring a numerical constant. For the first time, we distinguish the hardness of minimizing a PL function and a strongly convex function in terms of first-order complexities, as the momentum technique for smooth and strongly convex functions provably accelerates Gradient Descent by a certain polynomial order.

It is worth mentioning that the optimization problem under our consideration is high-dimensional and the goal is to obtain the complexity bounds that do not have an explicit dependency on the dimension.

Our technique to establish the lower bound follows from the previous lower bounds in convex (Nesterov, 2003) and non-convex optimization (Carmon et al., 2021). The main idea is to construct a so-called "zero-chain" function ensuring that any first-order algorithm per-iteratively can only solve one coordinate of the optimization variable. Then for a "zero-chain" function that has a sufficiently high dimension, some number of entries will never reach their optimal values after the execution of any first-order algorithm in certain iterations. To obtain the desired $\Omega\left(\frac{L}{\mu} \log \frac{1}{\varepsilon}\right)$ lower bound, we propose a "zero-chain" function similar to Carmon et al. (2020), which is composed of the worst convex function designed by Nesterov (2003) and a separable function in the form as $\sigma \sum_{i=1}^{T} v_{T,c}(\mathbf{x}_i)$ or $\sum_{i=1}^{T} v_{\mathbf{y}_i}(\mathbf{x}_i)$ to destroy the convexity. Different from their separable function, the one that we introduce has a large Lipshictz constant. This property helps us to estimate the PL constant in a convenient way. This new idea gives new insights into the constructions and analyses of instance functions, which might be potentially generalized to establish the lower bounds for other non-convex problems.

NOTATION

We use bold letters, such as $\mathbf{x}$, to denote vectors in the Euclidean space $\mathbb{R}^d$, and bold capital letters, such as $\mathbf{A}$, to denote matrices. $\mathbf{I}_d$ denotes the identity matrix of size $d \times d$. We omit the subscript and simply denote $\mathbf{I}$ as the identity matrix when the dimension is clear from context. For $\mathbf{x} \in \mathbb{R}^d$, we use $\mathbf{x}_i$ to denote its $i$th coordinate. We use $\operatorname{supp}(\mathbf{x})$ to denote the subscripts of non-zero entries of $\mathbf{x}$, i.e. $\operatorname{supp}(\mathbf{x}) = \{i : \mathbf{x}_i \neq 0\}$. We use $\operatorname{span}\left\{\mathbf{x}^{(1)}, \cdots, \mathbf{x}^{(n)}\right\}$ to denote the linear subspace spanned by $\mathbf{x}^{(1)}, \cdots, \mathbf{x}^{(n)}$, i.e. $\left\{\mathbf{y} : \mathbf{y} = \sum_{i=1}^{n} a_i \mathbf{x}^{(i)}, a_i \in \mathbb{R}\right\}$. We call a function $f$ $L$-smooth if $\nabla f$ is $L$-Lipschitz continuous, i.e. $\|\nabla f(\mathbf{x}) - \nabla f(\mathbf{y})\| \leq L\|\mathbf{x} - \mathbf{y}\|$. We denote $f^* = \inf_{\mathbf{x}} f(\mathbf{x})$.

We let $\mathbf{x}^*$ be any minimizer of $f$, i.e., $\mathbf{x}^* = \arg\min f$. We always assume the existence of $\mathbf{x}^*$. We say that $\mathbf{x}$ is an $\varepsilon$-approximate optimal point of $f$ when $f(\mathbf{x}) - f^* \leq \varepsilon$.

## 2 RELATED WORK

**Lower Bounds** There has been a line of research concerning the lower bounds of algorithms on certain function classes. To the best of our knowledge, (Nemirovskij & Yudin, 1983) defines the oracle model to measure the complexity of algorithms, and most existing research on lower bounds follow this formulation of complexity. For convex functions and first-order oracles, the lower bound is studied in Nesterov (2003), where well-known optimal lower bound $\Omega(\varepsilon^{-\frac{1}{2}})$ and $\Omega(\kappa \log \frac{1}{\varepsilon})$ are obtained. For convex functions and $n$th-order oracles, lower bounds $\Omega\left(\varepsilon^{-\frac{2}{3n+1}}\right)$ have been proposed in Arjevani et al. (2019b). When the function is non-convex, it is generally NP-hard to find its global minima, or to test whether a point is a local minimum or a saddle point (Murty & Kabadi, 1985). Instead of finding $\varepsilon$-approximate optimal points, an alternative measure is finding $\varepsilon$-stationary points where $\|\nabla f(\mathbf{x})\| \leq \varepsilon$. Sometimes, additional constraints on the Hessian matrices of second-order stationary points are needed. Results of this kind include Carmon et al. (2020; 2021); Fang et al. (2018); Zhou & Gu (2019); Arjevani et al. (2019a; 2020). Though a PL function may be non-convex, it is tractable to find an $\varepsilon$-approximate optimal point, as local minima of a PL function must be global minima. In this paper, we give the lower complexity bound for finding $\varepsilon$-approximate optimal points.

**PL Condition** The PL condition was introduced by Polyak (Polyak, 1963) and Łojasiewicz (Lojasiewicz, 1963) independently. Besides the PL condition, there are other relaxations of the strong convexity, including error bounds (Luo & Tseng, 1993), essential strong convexity (Liu et al., 2014), weak strong convexity (Necoara et al., 2019), restricted secant inequality (Zhang & Yin, 2013), and quadratic growth (Anitescu, 2000). Karimi et al. (2016) discussed the relationships between these conditions. All these relaxations implies the PL condition except for the quadratic growth, which implies that the PL condition is quite general. There are many other papers that study designing practical algorithms to optimize a PL objective function under different scenarios, for example, Bassily et al. (2018); Nouiehed et al. (2019); Hardt & Ma (2016); Fazel et al. (2018); J Reddi et al. (2016); Lei et al. (2017).

## 3 PRELIMINARIES

### 3.1 UPPER BOUND ON PL FUNCTIONS

Although the PL condition is a weaker condition than strong convexity, it guarantees linear convergence for Gradient Descent. The result can be found in Polyak (1963) and Karimi et al. (2016). We present it here for completeness.

**Theorem 1.** *If $f$ is $L$-smooth and satisfies $\mu$-PL condition, then the Gradient Descent algorithm with a constant step-size $\frac{1}{L}$:*

$$\mathbf{x}^{(k+1)} = \mathbf{x}^{(k)} - \frac{1}{L}\nabla f(\mathbf{x}^{(k)}), \tag{3}$$

*has a linear convergence rate. We have:*

$$f(\mathbf{x}^{(k)}) - f^* \leq \left(1 - \frac{\mu}{L}\right)^k (f(\mathbf{x}^{(0)}) - f^*). \tag{4}$$

Theorem 1 shows that the Gradient Descent algorithm finds the $\varepsilon$-approximate optimal point of $f$ in $O\left(\frac{L}{\mu} \log \frac{1}{\varepsilon}\right)$ gradient computations. This gives an upper complexity bound for first-order algorithms. However, it remains open to us whether there are faster algorithms for smooth PL functions. We will establish a lower complexity bound on first-order algorithms, which nearly matches the upper bound.

## 3.2 Definitions of algorithm classes and function classes

An algorithm is a mapping from real-valued functions to sequences. For algorithm A and $f : \mathbb{R}^d \to \mathbb{R}$, we define $\mathsf{A}[f] = \{\mathbf{x}^{(i)}\}_{i \in \mathbb{N}}$ to be the sequence of algorithm A acting on $f$, where $\mathbf{x}^{(i)} \in \mathbb{R}^d$.

Note here, the algorithm under our consideration works on function defined on any Euclidean space. We call it the dimension-free property of the algorithm.

The definition of algorithms abstracts away from the the optimization process of a function. We consider algorithms which only make use of the first-order information of the iteration sequence. We call them first-order algorithms. If an algorithm is a first-order algorithm, then

$$\mathbf{x}^{(i)} = \mathsf{A}^{(i)} \left( \mathbf{x}^{(0)}, \nabla f(\mathbf{x}^{(0)}), \cdots, \mathbf{x}^{(i-1)}, \nabla f(\mathbf{x}^{(i-1)}) \right), \tag{5}$$

where $\mathsf{A}^{(i)}$ is a function depending on A. Perhaps the simplest example of first-order algorithms is Gradient Descent.

We are interested in finding an $\varepsilon$-approximate point of a function $f$. Given a function $f : \mathbb{R}^d \to \mathbb{R}$ and an algorithm A, the complexity of A on $f$ is the number of queries to the first-order oracle needed to find an $\varepsilon$-approximate point. We denote $T_\varepsilon(\mathsf{A}, f)$ to be the gradient complexity of A on $f$, then

$$T_\varepsilon(\mathsf{A}, f) = \min_t \left\{ t : f(\mathbf{x}^{(t)}) - f^* \leq \varepsilon \right\}. \tag{6}$$

In practice, we do not have the full information of the function $f$. We only know that $f$ is in a particular function class $\mathcal{F}$, such as $L$-smooth functions. Given an algorithm A. We denote $\mathsf{T}_\varepsilon(\mathsf{A}, \mathcal{F})$ to be the complexity of A on $\mathcal{F}$, and define $\mathsf{T}_\varepsilon(\mathsf{A}, \mathcal{F})$ as follows:

$$\mathsf{T}_\varepsilon(\mathsf{A}, \mathcal{F}) = \sup_{f \in \mathcal{F}} T_\varepsilon(\mathsf{A}, f). \tag{7}$$

Thus, $\mathsf{T}_\varepsilon(\mathsf{A}, \mathcal{F})$ is the worst-case complexity of functions $f \in \mathcal{F}$.

For searching an $\varepsilon$-approximate optimal point of a function in $\mathcal{F}$, we need to find an algorithm which have a low complexity on $\mathcal{F}$. Denote an algorithm class by $\mathcal{A}$. The lower bound of an algorithm class on $\mathcal{F}$ describes the efficiency of algorithm class $\mathcal{A}$ on function class $\mathcal{F}$, which is defined to be

$$\mathcal{T}_\varepsilon(\mathcal{A}, \mathcal{F}) = \inf_{\mathsf{A} \in \mathcal{A}} \mathsf{T}_\varepsilon(\mathsf{A}, \mathcal{F}) = \inf_{\mathsf{A} \in \mathcal{A}} \sup_{f \in \mathcal{F}} T_\varepsilon(\mathsf{A}, f). \tag{8}$$

## 3.3 Zero-respecting Algorithm

Among all the algorithms, a special algorithm class is called zero-respecting algorithms. If A is a zero-respecting algorithm and $\mathsf{A}[f] = \{\mathbf{x}^{(t)}\}_{t \in \mathbb{N}}$, then the following condition holds for all $f : \mathbb{R}^d \to \mathbb{R}$:

$$\mathrm{supp}\{\mathbf{x}^{(n)} - \mathbf{x}^{(0)}\} \in \bigcup_{i=1}^{n-1} \mathrm{supp}\{\nabla f(\mathbf{x}^{(i)})\}. \tag{9}$$

That is to say, $\mathbf{x}^{(n)} - \mathbf{x}^{(0)}$ lies in the linear subspace spanned by $\nabla f(\mathbf{x}^{(0)}), \cdots, \nabla f(\mathbf{x}^{(n-1)})$. We denote the collection of first-order zero-respecting algorithms with $\mathbf{x}^{(0)} = \mathbf{0}$ by $\mathcal{A}_{\mathrm{zr}}$. It is shown by Nemirovskij & Yudin (1983) that a lower complexity bound on first-order zero-respecting algorithms are also a lower complexity bound on all the first-order algorithm when the function class satisfies the orthogonal invariance property.

## 3.4 Zero-chain

A zero-chain $f$ is a function that safisfies the following condition:

$$\mathrm{supp}(\mathbf{x}) \subseteq \{1, 2, \cdots, k\} \implies \mathrm{supp}(\nabla f(\mathbf{x})) \subseteq \{1, 2, \cdots, k+1\}, \quad \forall \mathbf{x}. \tag{10}$$

In other words, the support of $\nabla f(\mathbf{x})$ lies in a restricted linear subspace depending on the support of $\mathbf{x}$.

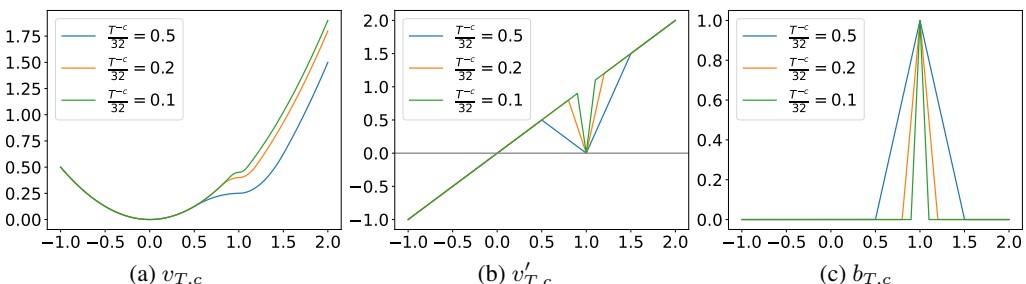

Figure 1: Illustration of $v_{T,c}$, $v'_{T,c}$ and $b_{T,c}$.

The "worst function in the (convex) world" in Nesterov (2003) defined as

$$f_k(\mathbf{x}) = \frac{1}{2}(\mathbf{x}_1 - 1)^2 + \sum_{i=1}^{k-1}(\mathbf{x}_{i+1} - \mathbf{x}_i)^2 \tag{11}$$

is a zero-chain, because if $\mathbf{x}_i = 0$ for $i > n$, then $(\nabla f_k(\mathbf{x}))_{i+1} = 0$ for $i > n$. A zero-chain is difficult to optimize for zero-respecting algorithms, because zero-respecting algorithms only discover one coordinate by one gradient computation.

## 4 MAIN RESULTS

According to Theorem 1, we already have an upper complexity bound $O\left(\frac{L}{\mu}\log\frac{1}{\varepsilon}\right)$ by applying Gradient Descent to all the PL functions. In this section, we establish the lower complexity bound of first-order algorithms on PL functions. Let $\mathcal{P}(\Delta, \mu, L)$ be the collection of all $L$-smooth and $\mu$-PL functions $f$ with $f(\mathbf{x}^{(0)}) - f^* \leq \Delta$. We establish a lower bound of $\mathcal{T}_\varepsilon(\mathcal{A}_{\mathrm{zr}}, \mathcal{P}(\Delta, \mu, L))$ by constructing a function which is hard to optimize for zero-respecting algorithms, and extend the result to first-order algorithms. We first propose a relatively simple hard instance which leads to a $\Omega\left(\kappa^{1-a}\right)$ lower bound in Section 4.1, where $a$ can be any real number that belongs to $(0, 1)$. This helps in understanding our intuitions. Then we present a more complicated hard instance that can achieve the desired $\Omega\left(\kappa\log\frac{1}{\varepsilon}\right)$ lower bound in Section 4.2.

### 4.1 $\Omega\left(\kappa^{1-a}\right)$ LOWER BOUND

We introduce a hard function $f_{T,c,\sigma}: \mathbb{R}^T \to \mathbb{R}$ for first-order algorithms:

$$f_{T,c,\sigma}(\mathbf{x}) = q(\mathbf{x}) + \sigma\sum_{i=1}^{T} v_{T,c}(\mathbf{x}_i), \tag{12}$$

where

$$q(\mathbf{x}) = \frac{1}{2}\mathbf{x}_1^2 + \frac{1}{2}\sum_{i=1}^{T-1}(\mathbf{x}_{i+1} - \mathbf{x}_i)^2 \tag{13}$$

is a quadratic function, and we define $v_{T,c}$ as follows:

$$v_{T,c}(x) = \begin{cases} \frac{1}{2}x^2, & x \leq 1 - \frac{1}{32}T^{-c}, \\ \frac{1}{2}x^2 - 16T^c\left(x - 1 + T^{-c}\right)^2, & 1 - \frac{1}{32}T^{-c} < x \leq 1, \\ \frac{1}{2}x^2 - \frac{1}{32}T^{-c} + 16T^c\left(x - 1 - T^{-c}\right)^2, & 1 < x \leq 1 + \frac{1}{32}T^{-c}, \\ \frac{1}{2}x^2 - \frac{1}{32}T^{-c}, & x > 1 + \frac{1}{32}T^{-c} \end{cases} \tag{14}$$

where $T$ is a positive integer, and $c, \sigma$ are positive real numbers.

The function $q$ can be rewritten as

$$q(\mathbf{x}) = \frac{1}{2}\mathbf{x}^T\mathbf{A}\mathbf{x}, \tag{15}$$

where

$$
\mathbf{A} = \begin{bmatrix} 2 & -1 & & & \\ -1 & 2 & -1 & & \\ & \ddots & \ddots & \ddots & \\ & & -1 & 2 & -1 \\ & & & -1 & 1 \end{bmatrix}
\tag{16}
$$

is a positive-definite symmetric matrix.

From the definition of $v_{T,c}$ in (14), we obtain the derivative of $v_{T,c}$, which is:

$$
v'_{T,c}(x) = \begin{cases} x, & x \le 1 - \frac{1}{32}T^{-c}, \\ x - 32T^c \left(x - 1 + \frac{1}{32}T^{-c}\right), & 1 - \frac{1}{32}T^{-c} < x \le 1, \\ x + 32T^c \left(x - 1 - \frac{1}{32}T^{-c}\right), & 1 < x \le 1 + \frac{1}{32}T^{-c}, \\ x, & x > 1 + \frac{1}{32}T^{-c}. \end{cases}
\tag{17}
$$

As we can see in (17), $v'_{T,c}$ is a piecewise linear function. To further simplify notations, we define $b_{T,c}(x)$ in (18):

$$
b_{T,c}(x) = \begin{cases} 1 - 32T^c |x - 1|, & 1 - \frac{1}{32}T^{-c} \le x \le 1 + \frac{1}{32}T^{-c}, \\ 0, & \text{otherwise.} \end{cases}
\tag{18}
$$

Then $v'_{T,c}(x) = x - b_{T,c}(x)$. Figure 1 provides geometric view of $v_{T,c}$ and $b_{T,c}$.

The quadratic part $q$ is a translation of "the worst function in the (convex) world" in Nesterov (2003), and the definition of $v_{T,c}$ is inspired by the hard instance in Carmon et al. (2021). Our hard instance differs from previous ones mainly in the large Lipschitz constant of its gradient. We note that the controlled degree of nonsmoothness is crucial for our estimate of PL constant. In Lemma 1 we list some important properties of $f_{T,c,\sigma}$, which we prove in Section **??**.

**Lemma 1.** *If $\sigma < 1$ and $T^c \ge \frac{1}{2}\sigma^{-1}$, then $f_{T,c,\sigma}$ satisfies the following.*

1. *$g_{T,c,\sigma}(\mathbf{x}) = f_{T,c,\sigma}(\mathbf{1} - \mathbf{x})$ is a zero-chain.*

2. *$\mathbf{x}^* = \mathbf{0}$, $f^*_{T,c,\sigma} = 0$, $f_{T,c,\sigma}(x) \le \frac{1}{2}\mathbf{x}^T(\mathbf{A} + \sigma\mathbf{I})\mathbf{x}$.*

3. *$f_{T,c,\sigma}$ is $(4 + \sigma + 32\sigma T^c)$-smooth.*

4. *$f_{T,c,\sigma}$ satisfies the $\frac{1}{C_1 T^{1+5c}}$-PL condition, where $C_1$ is a universal constant.*

Now we study the lower bound of zero-respecting algorithms first. Let $\tilde{f}$ denote the following function:

$$
\tilde{f}(\mathbf{x}) = \frac{LT^{-1}D^2}{42\sigma T^c} f_{T,c,\sigma}\left(\mathbf{1} - T^{1/2}D^{-1}\mathbf{x}\right),
\tag{19}
$$

where $D$ is the distance between $\mathbf{0}$ and $\mathbf{x}^*$, and $T$, $\sigma$ are parameters to be specified later. A change in $D$ affects $\tilde{f}(\mathbf{0}) - \tilde{f}(\mathbf{x}^*)$ and $\|\mathbf{x}^* - \mathbf{0}\|$, but does not affect the condition number of $\tilde{f}$.

In Lemma 2, we show that $\tilde{f}$ is difficult to optimize for all first-order zero-respecting algorithms.

**Lemma 2.** *If $\sigma < 1$ and $T^c > \frac{1}{2}\sigma^{-1}$, a first-order zero-respecting algorithm with $\mathbf{x}^{(0)} = \mathbf{0}$ needs at least $\frac{T}{2}$ gradient computations to find a point $\mathbf{x}$ satisfying $\tilde{f}(\mathbf{x}) - \tilde{f}^* \le \frac{1}{16}(\tilde{f}(\mathbf{x}^{(0)}) - \tilde{f}^*)$.*

We give the full proof of Lemma 2 in Appendix B.1 . The key point is that each gradient access of a zero-respecting algorithm only reveals one coordinate of a zero-chain. For $i > \frac{T}{2}$, $\mathbf{x}_i^{(k)}$ remains unchanged when $k \le \frac{T}{2}$, which gives a lower bound of the function value after the first $\frac{T}{2}$ gradient computations.

With Lemma 1 and Lemma 2 in hand, we are ready to give our lower complexity bound for zero-respecting first-order algorithms.

**Theorem 2.** *Given $L \geq \mu > 0$ and $c < 0.01$. When $\kappa = \frac{L}{\mu} > C_2$ where $C_2$ is a universal constant, we let*

$$T \in \left( \left( \frac{100}{C_2} \kappa \right)^{1/(1+6c)}, \left( \frac{200}{C_2} \kappa \right)^{1/(1+5c)} \right) \cap \mathbb{Z}, \tag{20}$$

*and*

$$\sigma = \frac{100}{C_2} \kappa T^{-(1+6c)}. \tag{21}$$

*Then, $\tilde{f}$ is $L$-smooth and $\mu$-PL. Moreover, any first-order zero-respecting algorithm with $\mathbf{x}^{(0)} = \mathbf{0}$ needs at least $\Omega\left(\kappa^{1/(1+6c)}\right)$ gradient computations to find a point $\mathbf{x}$ satisfying $\tilde{f}(\mathbf{x}) - \tilde{f}^* \leq \frac{1}{16}(\tilde{f}(\mathbf{x}^{(0)}) - \tilde{f}^*)$.*

The proof of Theorem 2 is provided in Section **??**. For $a$ satisfying $\frac{a}{6(1-a)} < 0.01$, let $c = \frac{a}{6(1-a)}$, then by Theorem 2, any zero-respecting algorithm needs at least $\Omega(\kappa^{1-a})$ gradient computations to find an $\varepsilon$-approximate optimal point of the function $\tilde{f}$.

Using the technique of Nemirovskij & Yudin (1983), for specific function classes such as PL functions, a lower complexity bound on first-order zero-respecting algorithms is also a lower complexity bound on all the first-order algorithms. Denoting the set of all first-order algorithms by $\mathcal{A}^{(1)}$, we have the following lemma:

**Lemma 3.**

$$\mathcal{T}_\varepsilon \left( \mathcal{A}^{(1)}, \mathcal{P}(\Delta, \mu, L) \right) \geq \mathcal{T}_\varepsilon \left( \mathcal{A}_{\mathrm{zr}}, \mathcal{P}(\Delta, \mu, L) \right). \tag{22}$$

Note that given $L$ and $\mu$, the value $\tilde{f}(\mathbf{0})$ can be controlled by choosing an appropriate $D$. Lemma 2 and 3 leads to the main result of the paper:

**Theorem 3.** *For any $0 < a < 1$, when $\varepsilon \leq \frac{1}{16}\Delta$,*

$$\mathcal{T}_\varepsilon \left( \mathcal{A}^{(1)}, \mathcal{P}(\Delta, \mu, L) \right) \geq \Omega(\kappa^{1-a}) \tag{23}$$

**Discussion about parameter setting** In the definition of $\tilde{f}$, $\tilde{f}$ has hyper-parameters $T$, $\sigma$, $c$. From our construction, to achieve a lower bound of $\Omega(\kappa^{1-a})$ when $a$ tends to zero, $c$ also tends to zero. Parameters $T$ and $\sigma$ are chosen according to (20) and (21). When $c$ tends to zero, $T = \Theta(\kappa)$ and $\sigma$ tends to 1.

## 4.2 $\Omega\left(\kappa \log \frac{1}{\varepsilon}\right)$ LOWER BOUND

We show that the $\Omega(\kappa)$ lower bound can be further improved to $\Omega\left(\kappa \log \frac{1}{\varepsilon}\right)$ with a new hard instance based on $f_{T,c,\sigma}$ and a similar technique to estimate the PL constant. Detailed proof of Lemmas and Theorems in this subsection is provided in Appendix C .

We first introduce several components of the new hard instance. We define

$$v_y(x) = \begin{cases} \frac{1}{2}x^2, & x \leq \frac{31}{32}y, \\ \frac{1}{2}x^2 - 16\left(x - \frac{31}{32}y\right)^2, & \frac{31}{32}y < x \leq y, \\ \frac{1}{2}x^2 - \frac{y^2}{32} + 16\left(x - \frac{33}{32}y\right)^2, & y < x \leq \frac{33}{32}y, \\ \frac{1}{2}x^2 - \frac{y^2}{32}, & x > \frac{33}{32}y, \end{cases} \tag{24}$$

where $y > 0$ is a constant. By the definition of $v_y$, we have

$$v_y'(x) = \begin{cases} x, & x \leq \frac{31}{32}y, \\ x - 32\left(x - \frac{31}{32}y\right), & \frac{31}{32}y < x \leq y, \\ x + 32\left(x - \frac{33}{32}y\right), & y < x \leq \frac{33}{32}y, \\ x, & x > \frac{33}{32}y. \end{cases} \tag{25}$$

For the non-convex part, we define

$$b_y(x) = \begin{cases} y - 32|x - y|, & \frac{31}{32}y \le x \le \frac{33}{32}y, \\ 0, & \text{otherwise.} \end{cases} \qquad (26)$$

Then we have $v'_y(x) = x - b_y(x)$.

For the convex part, we define $q_{T,t}(\mathbf{x})$ as follows (for the convenience of notation, we define $\mathbf{x}_0 = 0$):

$$q_{T,t}(\mathbf{x}) = \frac{1}{2} \sum_{i=0}^{t-1} \left[ \left( \frac{7}{8}\mathbf{x}_{iT} - \mathbf{x}_{iT+1} \right)^2 + \sum_{j=1}^{T-1} (\mathbf{x}_{iT+j+1} - \mathbf{x}_{iT+j})^2 \right], \qquad (27)$$

where $\mathbf{x} \in \mathbb{R}^{Tt}$. $q_{T,t}$ is a quadratic function of $\mathbf{x}$, thus can be written as

$$q_{T,t}(\mathbf{x}) = \frac{1}{2}\mathbf{x}^T\mathbf{B}\mathbf{x}, \qquad (28)$$

where $\mathbf{B}$ is a positive semi-definite symmetric matrix. Like $\mathbf{A}$, $\mathbf{B}$ also satisfies $0 \preceq \mathbf{B} \preceq 4\mathbf{I}$, because the sum of absolute value of non-zero entries of each row of $\mathbf{B}$ is smaller or equal to $4$.

Let $\mathbf{y} \in \mathbb{R}^{Tt}$ be a vector satisfying $\mathbf{y}_{qT+b} = \left( \frac{7}{8} \right)^q$, where $q \in \mathbb{N}$, $b \in \{1, 2, \cdots, T\}$. We define the hard instance $g_{T,t} : \mathbb{R}^{Tt} \to \mathbb{R}$ as follows:

$$g_{T,t}(\mathbf{x}) = q_{T,t}(\mathbf{x}) + \sum_{i=1}^{Tt} v_{\mathbf{y}_i}(\mathbf{x}_i). \qquad (29)$$

Now we list some properties of $g_{T,t}$ in Lemma 4.

**Lemma 4.** *$f_{T,c,\sigma}$ satisfies the following.*

1. *$g_{T,t}(\mathbf{y} - \mathbf{x})$ is a zero-chain.*

2. *$\mathbf{x}^* = \mathbf{0}$, $g_{T,t}^* = 0$, $g_{T,t}(\mathbf{x}) \le \frac{1}{2}\mathbf{x}^T(\mathbf{B} + \mathbf{I})\mathbf{x}$.*

3. *$g_{T,t}$ is 37-smooth.*

4. *$g_{T,t}$ satisfies the $\frac{1}{C_3 T}$-PL condition, where $C_3$ is a universal constant.*

Define $\tilde{g}$ to be the following function, which is hard for first-order algorithms:

$$\tilde{g}(\mathbf{x}) = \frac{LT^{-1}D^2}{37} g_{T,t}\left( \mathbf{y} - T^{1/2}D^{-1}\mathbf{x} \right), \qquad (30)$$

Similar to Lemma 2, we show that $\tilde{g}$ is hard for first-order zero-respecting algorithms:

**Lemma 5.** *Assume that $\varepsilon < 0.01$ and let $t = 2 \left\lfloor \log_{\frac{8}{7}} \frac{3}{2\varepsilon} \right\rfloor$. A first-order zero-respecting algorithm with $\mathbf{x}^{(0)} = \mathbf{0}$ needs at least $\frac{1}{2}Tt$ gradient computations to find a point $\mathbf{x}$ satisfying $\tilde{g}(\mathbf{x}) - \tilde{g}^* \le \varepsilon(\tilde{g}(\mathbf{x}^{(0)}) - \tilde{g}^*)$.*

With Lemma 4 and 5, we obtain a lower bound for zero-respecting algorithms:

**Theorem 4.** *Given $L \ge \mu > 0$. When $\kappa = \frac{L}{\mu} > C_4$ where $C_4$ is a universal constant, there exists $T$ and $t$ such that $\tilde{g}$ is $L$-smooth and $\mu$-PL. Moreover, any first-order zero-respecting algorithm with $\mathbf{x}^{(0)} = \mathbf{0}$ needs at least $\Omega\left( \kappa \log \frac{1}{\varepsilon} \right)$ gradient computations to find a point $\mathbf{x}$ satisfying $\tilde{g}(\mathbf{x}) - \tilde{g}^* \le \varepsilon(\tilde{g}(\mathbf{x}^{(0)}) - \tilde{g}^*)$.*

Finally, we arrive at a lower bound for first-order algorithms using Lemma 3:

**Theorem 5.** *For any $0 < a < 1$, when $\varepsilon \le \frac{1}{16}\Delta$,*

$$\mathcal{T}_\varepsilon\left( \mathcal{A}^{(1)}, \mathcal{P}(\Delta, \mu, L) \right) \ge \Omega\left( \kappa \log \frac{1}{\varepsilon} \right) \qquad (31)$$

This bound matches the convergence rate of Gradient Descent up to a constant.

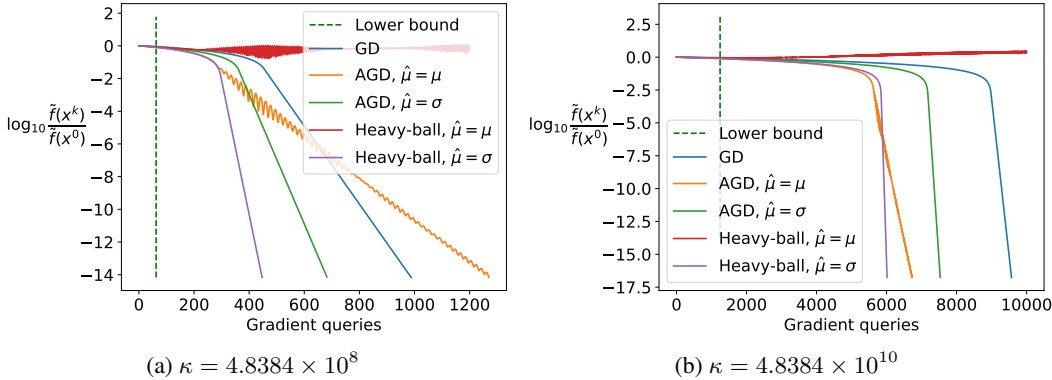

(a) $\kappa = 4.8384 \times 10^8$    (b) $\kappa = 4.8384 \times 10^{10}$

Figure 2: Convergence rate under Gradient Descent, Nesterov's Accelerated Gradient Descent and Polyak's Heavy-ball Method

## 5 NUMERICAL EXPERIMENTS

We conduct numerical experiments on our hard instance. We consider the $\kappa$ relatively large, which can reduce the factors from the numerical constants. We $\kappa = 4.8384 \times 10^8$, $c = 0.001$ in Figure 2a and $\kappa = 4.8384 \times 10^{10}$, $c = 0.001$ in Figure 2b, then compute corresponding $T$ and $\sigma$ according to (20) and (21). We use Gradient Descent, Nesterov's Accelerated Gradient Descent (AGD) and Polyak's Heavy-ball Method to optimize the hard instance. As AGD and the Heavy-ball Method are designed for convex functions, we need to choose appropriate parameter $\hat{\mu}$ in both algorithms, because our hard instance is non-convex. The first choice is $\mu$, which is the PL constant of our hard instance. The second choice is $\sigma$, because when $\mathbf{x}_i$ is far from 1, our hard instance can be treated as a $\sigma$-strongly convex quadratic function.

From Figure 2, we observe that the convergence can be roughly divided into two phases. In phase one, the optimization methods tries to pull every coordinate of $\tilde{\mathbf{x}}$ away from 1, and the function value does not decrease much. In phase two they converge linearly. This may be due to the fact that when each $\tilde{\mathbf{x}}_i$ is far from 1, our hard instance is exactly a strongly convex quadratic function. The observation is consistent with our theoretical results.

## 6 CONCLUSION

We construct a lower complexity bound on optimizing smooth PL functions with first-order methods. A first-order algorithm needs at least $\Omega\left(\frac{L}{\mu}\log\frac{1}{\varepsilon}\right)$ gradient access to find an $\varepsilon$-approximate optimal point of an $L$-smooth $\mu$-PL function. Our lower bound matches the convergence rate of Gradient Descent up to constants.

We only focus on deterministic algorithms in this paper. We conjecture that our results can be extended to randomized algorithms, using the same technique in Nemirovskij & Yudin (1983) and explicit construction in Woodworth & Srebro (2016) and Woodworth & Srebro (2017). We leave its formal derivation to the future work.

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
