# OpenReview forum: "On the Lower Bound of Minimizing Polyak-Łojasiewicz functions"
_ICLR.cc/2023/Conference — Submitted to ICLR 2023_

### Official Review · Reviewer_dAiB · 2022-10-25

**Confidence:** 4
**Correctness:** 4
**Technical Novelty And Significance:** 3
**Empirical Novelty And Significance:** Not applicable
**Recommendation:** 5

**Clarity, Quality, Novelty And Reproducibility:**

The paper is well-written, the flow is very clear. Also it is a rather concise paper without lengthy proof, appreciate it.

The main technique is still the zero-chain argument which is common in lower bound literature, but the proposed hard instance reveals novelty.

**Strength And Weaknesses:**

Strength:
1. First specific lower bound result for PL functions
2. Reveals the fundamental difference between PL function and strongly convex functions
3. The paper is well-written.

Weakness:
1. The lower bound comes without the dependence on $\epsilon$, which is different from the strongly convex counterpart.

**Summary Of The Paper:**

This paper studied the lower bound of minimizing PL functions. It provided an $\Omega((L/\mu)^{1-a})$ (arbitrary small $a>0$) lower bound for finding the optimal solution. It reveals a fundamental difference between PL functions and strongly convex minimization, also it shows that GD is already nearly optimal for solving PL functions.

**Summary Of The Review:**

To be honest, I am in a dilemma on evaluating the significance of the result. I am excited on the result concerning it is the first result for PL functions. But the lower bound here is just $\Omega((L/\mu)^{1-a})$, which is kind of "constant" level, rather than the $\Omega(\sqrt{\kappa}\log(1/\epsilon))$ in the strongly convex case, or maybe the lower bound here can be further rewritten as ($\kappa=L/\mu$):

$\Omega(\max(\sqrt{\kappa}\log(1/\epsilon), \kappa^{1-a}))$

So I believe that the result in the submission should not be tight enough (maybe the final tight lower bound should be $\Omega(\kappa\log(1/\epsilon))$?), which makes me skeptical on the significance of the result. So I hope to learn more from authors on the insight of the result, also the difference on the technical part compared to the lower bound of strongly convex minimization. Thank you for the effort.

---------Update---------

I appreciate authors' effort for revising the result, and there is a significant change in the new version, which took me a long time to read it beyond the regular review time frame. I think it is a pretty nice result revealing high importance, but I would say it is almost another new paper and I agree with Reviewer a1uz that it may require another full round of serious review, so I will keep my score here. Thank you very much.

---

> ### Author Response · Authors · 2022-11-18
> **Thank you for your review**
>
> - I believe that the result in the submission should not be tight enough (maybe the final tight lower bound should be $\Omega(\kappa \log⁡(1/\epsilon))$?), which makes me skeptical on the significance of the result. So I hope to learn more from authors on the insight of the result, also the difference on the technical part compared to the lower bound of strongly convex minimization. Thank you for the effort.
>
> Thanks for your helpful suggestion on the term of $\epsilon$. In the revised paper, we have presented a new hard instance, which improves the lower bound to **$\Omega\left(\kappa\log\frac{1}{\epsilon}\right)$** as you have expected, matching the convergence rate of Gradient Descent up to a constant. The main result is presented in Section 4.3 and the proof is presented in Appendix C. The new hard instance can be seen as a concatenation of $\log\frac{1}{\epsilon}$ rescaled old hard instances. The estimate of PL constant follows from a similar technique, but needs a more careful analysis to remove the $\log\frac{1}{\epsilon}$ term.

---

### Official Review · Reviewer_sXZK · 2022-10-26

**Confidence:** 4
**Correctness:** 3
**Technical Novelty And Significance:** 2
**Empirical Novelty And Significance:** Not applicable
**Recommendation:** 8

**Clarity, Quality, Novelty And Reproducibility:**

I think the paper is well-written and is of good quality. The constructions, though inspired from prior works had to be modified in order to be applied in this context. Reproducibility: Does not apply.

**Strength And Weaknesses:**

Strengths:

- I think the problem is very natural and it was surprising (to me) that a result of this sort was not known. Also, the finding that gradient descent is optimal is interesting as well. This is because in (some parts) of the literature, PL functions are basically used as a class of non-convex functions in which typical analysis with strong convexity applies. However, this paper shows that unlike smooth strongly convex functions, it is not possible to accelerate gradient descent for this class. This establishes an interesting separation between the two classes.

- I think the paper is written well; it presents a good overview of the existing tools in lower bound constructions and gradually builds up towards the main result. Also, the main text contains sufficient details of the proof.

Weaknesses:
I think the main weakness is that the scope of the paper is rather limited. Moreover, I think that there are still some loose
ends which would have been good to resolve (or comment on its difficulties) given the limited scope. One such aspect is that the lower bound is only established for constant error -- while this is sufficient to argue that no acceleration is possible in this class; it would be good to show that the oracle complexity of gradient descent is tight even with regard to its dependence on $\epsilon$. Another aspect, unclear to me, is whether these lower bounds extend to randomized methods? If yes, the authors should comment on that; if not, it would be good to incorporate it (perhaps using techniques of [WS17]) or discuss challenges towards it.

[WS17]:  Lower bound for randomized first order convex optimization

I read parts of the proofs and have corrections and questions about how some of the steps follow. It would be helfpul if the authors response help address these.

- Don't understand why the inequality before Eqn. 53 holds. If the lower bound on $m$ in the statement is used, then there should be no $m$ in the right hand side.

- Proof of Lemma 5: I am little confused about the first few lines in the proof of this lemma. What if all coordinates of $\tilde x$ except the first (which is 0) are negative. Then the assertion that there exists $k,l$ such that $\tilde x_i>0.5$ for $i=k..l$ is not true. It seems that the proof should thus use the absolute values of these.

- I am not sure about what happened in inequality in Eqn. 59.

- Statement of Lemma 6: It seems that the assumptions stated in the lemma statement is not sufficient to get the final result -- say $x_l$ is $-\infty$, $x_{l+1} = 1- \frac{1}{32}T^{-c}$ and $x_{l+2} = 1+\frac{1}{32}T^{-c}$, then $(\nabla f_{T,c,\sigma} (x)){(l+1)} = -\infty$ The premise thus needs to be strengthened. Also, I dont understand some parts of the proof here:
(a). Third inequality -- how? what if $\tilde x_{l+1}$ is negative?
(b). What happened in the fourth inequality; the statement has no lower bound on $\tilde x_l$ in terms of $\tilde x_{l+1}$, right?

- Typo: It seems $1/32$ missing from definition $v_{T,c}$?

- Typo: $\sigma$ missing in front of $b_{T,c}(x)$ in inequality after Eqn. 66 and henceforth.

- Typo: after Eqn 70, inequality should be outside the norm.

- Typo: should be $1+\frac{T^{-c}}{32}$ in the rhs in the first sentence of part 4(c) of proof of Lemma 1.

- Typo: Should be $\tilde x_k$ (as opposed to $\tilde x_m$) at the end of base case in proof of Lemma 5

- Typo: In the proof of Lemma 2, the last inequality should be $\geq$.

**Summary Of The Paper:**

The paper presents a lower bound on gradient oracle complexity for optimizing smooth PL functions. Their result establishes that gradient descent method is worst-case optimal for this class of functions.

**Summary Of The Review:**

I think the studies a very natural problem and provides an interesting result. The only downside is its limited scope.

Edit: Post author response, I have increased my score from 6 to 8.

---

> ### Author Response · Authors · 2022-11-18
> **Thank you for your review**
>
> Thanks for your careful reading and helpful suggestions. We have corrected all the typos which you have pointed out. Now we clarify some technical details.
>
> - Don't understand why the inequality before Eqn. 53 holds. If the lower bound on m in the statement is used, then there should be no m in the right hand side.
>
> Sorry for the confusion. In eqn. (53) (now eqn. (63)), there is an additional $m$ because as $m$ increases, $\frac{m-1}{m}$ also increases. So we have $\frac{m-1}{m}\ge \frac{\frac{1}{32}T^{-c}}{1+\frac{1}{32}T^{-c}}\ge \frac{T^{-c}}{48}$. The explaination is added in the revised paper.
>
>
> - Proof of Lemma 5: I am little confused about the first few lines in the proof of this lemma. What if all coordinates of $x$ except the first (which is 0) are negative. Then the assertion that there exists k,l such that $x_i>0.5$ for $i=k..l$ is not true. It seems that the proof should thus use the absolute values of these.
>
> About the proof of original Lemma 5, now Lemma 7. Thanks for pointing out an omitted case. Now we have added the case of $\tilde x_{l}<0.5$. In fact, this case is relatively easy to prove, because in this case $\tilde x_{l+1}-\tilde x_l$ is very large. See
>
>
> - I am not sure about what happened in inequality in Eqn. 59.
>
> Sorry for the confusion. There was a typo in the second line of original eqn. (59) (now eqn. (69)), where $(1+\sigma)$ should be $(2+\sigma)$. Detailed derivation of every inequality have been added in the revised paper, and we present it here: the first inequality is due to (67): $|-\tilde x_{i-1}+(2+\sigma)\tilde x_i-\tilde x_{i+1}|<\frac{\sigma}{4}$, and the second inequality is due to $\tilde x_n\ge (1+\frac\sigma2)\tilde x_{n-1}$ and $\tilde x_n\ge \frac12$.
>
> - Statement of Lemma 6: It seems that the assumptions stated in the lemma statement is not sufficient to get the final result -- say $x_l$ is $-\infty$, $x_{l+1}=1−\frac{1}{32}T^{−c}$ and $x_{l+2}=1+\frac{1}{32}T^{−c}$, then $(\nabla f_{T,c,\sigma}(x))(l+1)=−\infty$ The premise thus needs to be strengthened. Also, I dont understand some parts of the proof here: (a). Third inequality -- how? what if $\tilde x_{l+1}$ is negative? (b). What happened in the fourth inequality; the statement has no lower bound on $\tilde x_l$ in terms of $\tilde x_{l+1}$ right?
>
> Sorry for the confusion. The conditions are sufficient to derive the result. There is a typo in the second line of original eqn. (61) (now eqn. (71)), where $\tilde x_l$ should be $-\tilde x_l$. In your special case where $x_l=-\infty$, $x_{l+1}=1-\frac{1}{32}T^{-c}$ and $x_{l+2}=1+\frac{1}{32}T^{-c}$, $(\nabla f_{T,c,\sigma}(x))_{l+1}=+\infty$, as $(\nabla f_{T,c,\sigma}(x))_{l+1}\ge -x_l+2x_{l+1}-x_{l+2}$. For line 3, since we have assumed that $\tilde x_{l+1}\ge 1-\frac{1}{32}T^{-c}$, so it cannot be negative. For line 4, we have assumed that $x_{l+1}\ge (1+\frac\sigma 2)x_l$, which is a lower bound on $-\tilde x_l$ in terms of $\tilde x_{l+1}$.
>
> About the weaknesses:
>
> - One such aspect is that the lower bound is only established for constant error -- while this is sufficient to argue that no acceleration is possible in this class; it would be good to show that the oracle complexity of gradient descent is tight even with regard to its dependence on \epsilon.
>
> Much thanks for your suggestion, which encouraged us to work on constructing a new lower bound. We have extended our result to a lower bound of $\Omega\left(\kappa\log\frac{1}{\epsilon}\right)$. The result is presented in Section 4.3 and the full proof is presented in Appendix C.  The proof follows from a similar technique to the original proof, but needs a more careful analysis to remove the $\log \frac{1}{\epsilon}$ in the PL constant.
>
> - Another aspect, unclear to me, is whether these lower bounds extend to randomized methods? If yes, the authors should comment on that; if not, it would be good to incorporate it (perhaps using techniques of [WS17]) or discuss challenges towards it.
>
> - [WS17]: Lower bound for randomized first order convex optimization
>
> Thanks for your suggestion. We have gone through [WS17].  We think our lower bound might be extended to randomized algorithm, using the same technique in [WS17].  Becuase we can also similarily  construct a distribution over high-dimensional hard instances. However, due to the limited time, we do not have a complete proof up to now.  We will work on it in the near future.

---

> > ### Comment · Reviewer_sXZK · 2022-11-21
> > **Thank you**
> >
> > I thank the authors for their detailed response and particulary for proving the improved lower bound which has the correct dependence on $\epsilon$. I also thank the authors for going through WS17 and their comment that their lower bound can be extended to randomized methods too; I think this would indeed be a solid addition to the paper (if it indeed turns out to be a *simple* extension). Regardless, I think that the improved lower bound itself further strengthens the result in the paper and so therefore I have raised my score to 8.

---

### Official Review · Reviewer_a1uz · 2022-10-28

**Confidence:** 4
**Correctness:** 3
**Technical Novelty And Significance:** 4
**Empirical Novelty And Significance:** Not applicable
**Recommendation:** 3

**Clarity, Quality, Novelty And Reproducibility:**

**Clarity.** Up to the end of Section 4.1 the paper is clearly written, though there are some minor issues with English. However, the proofs are hard to read due to many unexplained parts.

**Quality.** The quality of the paper is good given the strengths of the result (if the inaccuracies in the proofs are not fatal). Nevertheless, I encourage the authors to apply the necessary corrections. It will increase the quality a lot.

**Novelty.** The results are novel.

**Reproducibility.** Not applicable.

**Strength And Weaknesses:**

## Strengths

1. **Long-standing open question is resolved.** The contribution of the paper is fundamental: the question about the lower bounds for smooth PL functions has been open for a long time. PL condition gained a lot of interest in the optimization and ML communities during the recent few years, but the question remained open. This indicates the non-triviality of the derived result.

2. **Elegant worst-case example.** I find the constructed "worst-case" function quite elegant: it is a sum of Nesterov's "worst function in the world" function and piece-wise quadratic function (along each component). The first part is standard and provides nice properties such as zero-chain property, which is the key property of literally all existing lower bounds. The second part is non-standard and makes the problem non-convex but not too much, i.e., PL-condition holds. Therefore, the proposed worst-case function is very intuitive and quite simple. I believe that providing simple solutions (especially for old problems like the one that this work addresses) is highly valuable for the community.

## Weaknesses

1. **Inaccuracies and gaps in the proofs.** Unfortunately, the proofs are hard to read due to multiple inaccuracies and gaps. Most of these issues can be fixed (I have double-checked), but a few of them are not obvious to me (though I did not try to think about them too long -- this is the authors' responsibility to make the proofs as clear as possible). See them in the list of comments.

2. **Issues with English.** Some sentences end abruptly and some parts of the text are not polished well (especially in the proofs; because of that some parts are hard to comprehend). I provide some comments related to this issue. I suggest the authors make a thorough pass through the paper and correct all issues with English.

## Detailed comments

1. **Numerical experiment request.** The main result of this work is more than sufficient for publishing at top optimization/ML conference/journal (if the inaccuracies mentioned below are not fatal mistakes). However, I am curious to see how accelerated methods like Nesterov's one or momentum methods like the Heavy-ball method of Polyak behave on the constructed worst-case smooth PL function. It would be great to see the comparison of these methods with GD and with the theoretical lower bound for different dimensionalities of the problem.

2. "The PL condition may be originally introduced by Polyak": if I am not mistaken it was introduced by Polyak and Łojasiewicz independently.

3. "any $\hat \mu$-strongly convex function that admits": this sentence is a bit confusing because during the first reading one can think that the authors mean a particular subset of strongly convex functions. However, after this sentence, they provide a definition of differentiable strongly convex function. To prevent confusion, I suggest removing "that admits" (or replacing it by ", which by definition satisfies").

4. **Discussion about $\alpha, T, c$.** I believe that this discussion should be added somewhere in the text. In particular, $\alpha$ can be chosen arbitrarily small. However, in this case, $c$ has to be also small. This implies that $T$ should be large enough (according to (31) and (32)). In particular, $T \to \infty$ when $\alpha \to 0$. Therefore, the dimensionality of the worst-case example depends on $\alpha$. I believe, the paper will benefit from such kind of discussion.

5. Section 2 requires proofreading in terms of English: some sentences end abruptly.

6. The fonts in Figure 1 are too small. The authors should increase the fonts and also add markers to the curves (to make the plots easier to comprehend).

7. Formula (14) appears between two sentences. It should be a part of a sentence, e.g., "Next, we define $v_{T,c}$ as follows:"

8. Formula (14), the first row (condition on $x$), the second row (last term), the third row (the last term): one should add $\frac{1}{32}$ in front of $T^{-c}$.

9. Formula (19): what is the role of $D$? This parameter is never introduced.

10. Page 7, "for any $0 < a < 1$": one should also add that $\frac{a}{6(a-1)} < 0.01$ (otherwise the the condition on $c$ from Theorem 2 does not hold).

11. Proof sketch of Lemma 1: I think, this proof sketch does not improve the understanding of the proof. I believe it would be better to provide just the full proof of Lemma 1: for me it was easier to check the full proof than to parse the sketch.

12. Pages 8-9: "the smooth constant" --> "the smoothness constant"

13. Lemma 2, page 11: it is worth saying in the statement of the lemma that $T^{c} \geq \frac{1}{2}\sigma^{-1}$ and $\sigma < 1$.

14. Formula (43): the union is over $k$, not over $i$

15. Page 12, "Lemma 3 then follows from Proposition 2 in Carmon et al. (2020)": I guess one needs Proposition 1 to get the result. Moreover, it is formally about the slightly larger class of methods (so-called zero-respecting algorithms). As Carmon et al. (2020) write, this class is strictly larger than the class of the linear-span methods. However, the proof of Lemma 2 is valid even for zero-respecting algorithms, so, one can replace everywhere in the text "linear-span methods" by "zero-respecting methods" (and introduce zero-respecting methods instead of the linear-span ones).

16. Formula (48): $E_{T,T}$ is not defined. I guess it is $T\times T$ matrix having zero elements everywhere except the element $(T,T)$ that is equal to $1$.

17. After (49): I guess $e^{(i)}$ is a column of $I$, not a row (to have the right size).

18. After (50): one should exclude $k = 0$.

19. After (52), "From $x_k = \max |x_i|$": this sound a bit informal. Better to change to: "For $k$ such that $x_k = \max |x_i|$..."

20. Formula (53): in the fourth step one can get $\frac{1}{32}$ instead of $\frac{1}{64}$. Also the constant in the end should be different since $A + \sigma I \preceq (4+\sigma) I$ (see my comment 31).

21. Above (57): why $x_1$ cannot be larger than $1 - \frac{1}{32}T^{-c}$? It seems that the current proof requires $x_1$ to be smaller than $1 - \frac{1}{32}T^{-c}$.

22. Formula (57), the last step: if my calculations are correct, one can get $\sigma^2$ in the end (which is slightly better).

23. "The last inequality above holds because $||(A + \sigma I)x|| \leq 6\sqrt{T}$": my derivations give slightly better bound ($4$ instead of $6$). Moreover, the authors should provide the full derivation since it is not the readers' responsibility to fill the gaps in the proofs.

24. Formula (58): the authors should provide the complete derivation. I have double-checked this part myself, everything seems to be correct. However, as I already mentioned, this is the authors' responsibility to provide complete and clear proof.

25. Above (59), "If (56) holds for $i = n$": this is not a sentence. The authors should reformulate this part.

26. The last step of (59): I do not understand the last step. In particular, it is equivalent to $- \frac{1}{1 + \sigma/2}\tilde{x}_n \geq 0$. But this is not true. **I think this is an important gap in the proof.**

27. Formula (61): I do not understand how the fourth and sixth inequalities were obtained. Complete derivations are required. **I think this is an important gap in the proof.** Also I guess $\theta$ should be replaced by $\sigma$. Next, the final step also requires the full derivation (though this particular step is correct).

28. Formular (63): $b_{T,c} \to b_{T,c}(x)$.

29. The sentence after (63): I think it is better to add some details, e.g., $b_{T,c}(1) = 1$, which implies that $(\nabla g_{T,c,\sigma}(x))_j = (A(1-x))_j$ for $j \geq i$.

30. Formula (65): one should have factor $1 + 32T^c$, since $b_{T,c}$ does not depend on $\sigma$.

31. "Consequently, $f_{T,c,\sigma}$ is $(3 + 32\sigma T c)$-Lipschitz because $A \preceq 2I$": in fact, $A$ has eigenvalues larger than $3$, so, $A \preceq 2I$ does not hold. Instead, one can get $A \preceq 4I$ implying that $f_{T,c,\sigma}$ is $(4 + \sigma + 32\sigma T c)$-smooth (not Lipschitz)

32. Formula (66), the last step: one should add $\inf_{x\in \mathbb{R}^T}$ in front of the expression and also multiply $b_{T,c}(x)$ by $\sigma$.

33. Formula (68): I have double-checked this part, but the authors should provide a complete derivation. Also above (68) and everywhere in this proof one should replace $\max_{i=0,\ldots,T+1}\tilde{x}_i$ with $\max_{i=0,\ldots,T+1}|\tilde{x}_i|$.

34. Formula (69): I have double-checked this part, but the authors should provide a complete derivation.

35. After (69), (c): one should have $1 - \frac{1}{32} T^{-c} \leq \max_{i=0,\ldots,T+1}|\tilde{x}_i| \leq 1 + \frac{1}{32} T^{-c}$.

36. After (70), "consequently ...": the norm sign is in the wrong place.

37. Formula (71), the second step: according to my derivations, one should have $5$ instead of $3$ in the denominator of the second term. The authors should provide the complete derivation.

**Summary Of The Paper:**

This work derives a new lower bound for the class of first-order methods applied to minimize $L$-smooth functions satisfying the Polyak-Łojasiewicz condition with parameter $\mu$. In particular, the authors derive $\Omega\left(\left(\frac{L}{\mu}\right)^{1 - \alpha}\right)$ lower bound for finding $\varepsilon$-solution (for small enough $\varepsilon$ and large enough dimension of the problem that depends on $\alpha$). This closes a long-standing open question about the optimality of Gradient Descent for this class of problem.

**Summary Of The Review:**

Overall, the main result of the paper is very good and significant, if the mentioned inaccuracies are not fatal mistakes. I will increase the score if the authors fix at least the issues with the proofs (otherwise, I will have to decrease the score and recommend rejection). Moreover, the paper will benefit, if the authors provide the results of the numerical experiments requested in my review.



**UPDATE after rebuttal:**

I thank the authors for their effort and improvement of the lower bound. However, a lot of new content was added: original submission was 15 pages (with appendix), not it has 26 pages. These extra pages are quite technical and I can say that the changes are too significant. Therefore, the paper needs another full round of review. Unfortunately, I have to take this into account, that is why I am changing my score to 3.

---

> ### Author Response · Authors · 2022-11-18
> **Thank you for your comment**
>
> $\newcommand{\tx}{\tilde x}\newcommand{\tT}{\frac1{32}T^{-c}}\newcommand{\s}{\sigma}$
> Thank you for the detailed review and helpful comments. Following your suggestion, we work on the organization, add additional remarks and clarifications to improve the readability, and correct for typos and confusing notations. We address the technical comments below.
>
> - 1. **Numerical experiment request**.
>
> Thanks for your suggestion. Numerical experiments are added in the revised version in Section 5. We conducted numerical experiment on the hard instance using Gradient Descent, Nesterov's AGD and Polyak's Heavy-ball method. The observation is consistent with our theories.
>
> - 2. "The PL condition may be originally introduced by Polyak": it was introduced by Polyak and Łojasiewicz independently.
>
> Thanks for your suggestions. We modified the Introduction and Related work section in the paper to express that PL condition was introduced by Polyak and Łojasiewicz independently.
>
> - 3. "any $\hat\mu$-strongly convex function that admits": To prevent confusion, I suggest removing "that admits" (or replacing it by ", which by definition satisfies").
>
> We replaced "that admits" by "which by definition satisfies" before the definition of $\hat\mu$-strongly convex function.
>
> - 4. **Discussion about $\alpha,T,c$**.
>
> Thanks for your suggestion. We have added an discussion about $T$, $\s$ and $c$. Please see the end of Section 4.2 for details. As $c\to0$, $T\to\Theta(\kappa)$ and $\sigma\to1$.
>
> - 5, 7, 12, 19, 25. Thanks for your suggestion. We have carefully revised the English expressions to make it clean and readable.
>
> - 6. The fonts in Figure 1 are too small.
>
> We have adjusted the fonts in Figure 1 to make it more comprehendable
>
> - 8, 10, 14, 16, 17, 18, 28, 30, 32, 33, 35, 36.
>
> Thanks for pointing out these typos in our original paper. We have carefully revised our paper to reduce typos.
>
> - 9. Formula (19): what is the role of $D$?
>
> Thanks for your suggestions. The role of $D$ is the distance of $x_0$ and $x^*$. We have introduced its meaning after the definition of $\tilde f$.
>
> - 10. Page 7, "for any $0<a<1$": one should also add that $\frac{a}{6(a-1)}<0.01$(otherwise the the condition on $c$ from Theorem 2 does not hold).
>
> Thanks for pointing out this. We have added the condition in the revised paper.
>
> - 11.  Proof sketch of Lemma 1: I believe it would be better to provide just the full proof of Lemma 1.
>
> Thanks for your suggestions. We have removed the proof sketch of Lemma 1, and provided the full proof of Lemma 1 in Appendix B.3.
>
> - 13. Lemma 2, page 11: it is worth saying in the statement of the lemma that $T^c>\frac12\sigma^{-1}$ and $\sigma<1$.
>
> Thanks for your suggestion. We have added the statement in Lemma 2.
>
> - 15. One can replace "linear-span algorithm" with "zero-respecting algorithm".
>
> Thanks for your suggestions. If we let $x_0=0$, linear-span algorithm is a special case of zero-respecting algorithm. In the revised paper, we replace "linear-span algorithm" with "zero-respecting algorithm".
>
> - 20, 22. About constants.
>
> Thanks for the suggestions on the constants. We have changed some constants in the proof to make the proof consistent. This paper mainly focus on the term of $\kappa$. It is possible to improve the constants. However, obtaining a lower bound with a very tight constant is beyond the scope of the paper.
>
> - 21. Above (57): why $x_1$ cannot be larger than $1-\tT$?
>
> Thanks for pointing out an omitted case, where $x_l<0.5$ and $x_{l+1}>1-\tT$. When $x_1>1-\tT$, we may choose $k=l=0$, then the original (55) and (56) (now (65) and (66)) holds. The explanation has been added in the proof before (67).
>
> - 23, 24, 26, 27, 29, 33, 34. Full derivation of proofs.
>
> Thanks for your suggestions, and sorry for the confusion. We have provided full derivation of all the proofs in the revised paper.
>
> - 26. I do not understand the last step of (59). **I think this is an important gap in the proof**.
>
> Sorry for the confusion. There was an typo in the original version. The second line of original (59) (now (69)) should be $\ge(2+\s)\tx_n-\frac1{1+\frac{1}{2}}\tx_n\cdots$.
>
> We provide the full derivation of (69) here: the first inequality is due to (67), and the second inequality is due to $\tx_n\ge(1+\frac\s2)\tx_{n-1}$ and $\tx_n\ge\frac12$.
>
> - 27. Formula (61): I do not understand how the fourth and sixth inequalities were obtained. Complete derivations are required. **I think this is an important gap in the proof**.
>
> Sorry for the confusion. In original (61) (now (71)),The fourth inequality holds because $\tx_{l+1}\ge(1+\frac{\s}{2})\tx_l$ and $\tx_{l+2}\le1+\tT$. There was a typo in the sixth line. We omitted an $-1$ in the original version of the paper. The sixth line is due to $\tx_{l+1}\ge1-\tT$.
>
> - 31, 37. $f_{T,c,\s}$ is $(4+\sigma+32\s T^c)$-smooth
>
> Thanks for pointing out the issue in computing the smoothness constant of $f_{T,c,\s}$. We have corrected them in Lemma 1 of the revised version.

---

> > ### Comment · Reviewer_a1uz · 2022-11-25
> > **The paper was changed significantly. Another full round of review is needed**
> >
> > I thank the authors for their effort and improvement of the lower bound. However, a lot of new content was added: original submission was 15 pages (with appendix), not it has 26 pages. These extra pages are quite technical and I can say that the changes are too significant. Therefore, the paper needs another full round of review. Unfortunately, I have to take this into account, that is why I am changing my score to 3.

---

### Decision · Program_Chairs · 2023-01-20

**Decision:**

Reject

**Justification For Why Not Higher Score:**

Technical issues in the proofs.

**Justification For Why Not Lower Score:**

N/A

**Metareview: Summary, Strengths And Weaknesses:**

The authors derive a new lower bound for first-order methods applied to functions satisfying the PL (gradient dominance) condition. While I find lower bound results important and significant, I do not believe the paper could be published in its current form. This is due to some of the reviewers catching technical issues in the proofs. Although the authors have tried to address this, I believe the paper should go through a rigorous review process rather than a short rebuttal type of patching the technical shortcomings.